# Plasma from patients with pulmonary embolism show aggregates that reduce after anticoagulation

Stephen R. Baker [1,2], Georgia Halliday[1], Michal Ząbczyk[3], Ghadir Alkarithi[1,4], Fraser L. Macrae[1], Anetta Undas[3,5], Beverley J. Hunt[6] & Robert A. S. Ariëns [1✉]

## Abstract

**Background** Microclots, a term also used for amyloid fibrin(ogen) particles and henceforth named aggregates, have recently been reported in the plasma of patients with COVID-19 and long COVID. These aggregates have been implicated in the thrombotic complications of these diseases.

**Methods** Plasma samples from 35 patients with acute pulmonary embolism were collected and analysed by laser scanning confocal microscopy and scanning electron microscopy before and after clotting.

**Results** Here we confirm the presence of aggregates and show that they also occur in the plasma of patients with pulmonary embolism, both before and after clotting. Aggregates vary in size and consist of fibrin and platelets. We show that treatment with low-molecular weight heparin reduces aggregates in the samples of patients with pulmonary embolism. Double centrifugation of plasma does not eliminate the aggregates.

**Conclusions** These data corroborate the existence of microclots or aggregates in diseases associated with venous thromboembolism. Important questions are raised regarding their pathophysiological relevance and further studies are warranted to investigate whether they represent cause or consequence of clinical thrombosis.

## Plain language summary

When blood turns from liquid to solid, a protein called fibrin and cells called platelets aggregate to form a blood clot. Small aggregates have been found in the blood of people with COVID-19 and long COVID. Here, we show that small aggregates also occur in the blood of patients with pulmonary embolism, a disorder in which blood clots are trapped in an artery in the lung, preventing blood flow. We confirm that aggregates consist of fibrin and platelets, and show that the number of aggregates is lower when patients are treated with blood thinning drugs. These results suggest other disorders of the blood should also be investigated to see whether aggregates are present and whether they have an impact on the outcome for the patient. This could help us understand the cause of diseases associated with blood clotting, which might offer new approaches for diagnosis and treatment.

[1] Discovery and Translational Science Department, Leeds Institute of Cardiovascular and Metabolic Medicine, University of Leeds, Leeds, UK. [2] Department of Physics, Wake Forest University, Winston Salem, NC 27109, USA. [3] Institute of Cardiology, Jagiellonian University Medical College, Krakow, Poland. [4] Department of Medical Laboratory Technology, Faculty of Applied Medical Sciences, King Abdulaziz University, Jeddah, Saudi Arabia. [5] Krakow Center for Medical Research and Technologies, John Paul II Hospital, Krakow, Poland. [6] Thrombosis & Haemophilia Centre, Guys & St Thomas' NHS Foundation Trust, London, UK. ✉email: r.a.s.ariens@leeds.ac.uk

Coronavirus disease 2019 (COVID-19) is a respiratory disease caused by infection with severe acute respiratory syndrome coronavirus 2 (SARS-CoV-2), which has been associated with significant inflammation-driven abnormalities in coagulation and platelet activities leading to an increased risk of thrombosis[1]. Recently, microclots (from here-on termed aggregates) have been described in plasma samples from patients with type 2 diabetes mellitus[2], COVID-19[3], and patients with Long COVID/post-acute sequelae of COVID-19 (PASC)[4]. Aggregates staining positive for amyloid using thioflavin T were increased in a study of 33 patients with type 2 diabetes compared to 34 healthy controls ($2.52\% \pm 1.16\%$ vs $1.30 \pm 0.61\%$)[2], and this was further increased to an average level of 12–13% amyloid positive staining in another study of 20 patients with COVID-19[3]. The recent finding of aggregates in Long COVID has sparked a debate about their role in pathophysiology of disease[4].

Aggregates have been suggested by some to contribute to the thrombotic complications associated with these pathologies. However, the nature of the aggregates is not yet fully characterised, although they have been suggested to contain fibrin, or an amyloid form of fibrin[2,3], platelets[4], and a range of other plasma proteins including von Willebrand Factor, complement factors, C-reactive protein, factor XIII, plasminogen, α2-antiplasmin and serum amyloid protein[4]. Furthermore, all reports on plasma aggregates have thus far originated from one laboratory[2–4], and independent replication is hitherto lacking. Moreover, it is not clear how many of the patients present with plasma aggregates, whether aggregates are an in-vitro artefact of blood sampling or processing, and/or whether aggregates are cause or effect of active disease.

In this study, our aim was to (1) provide independent confirmation of aggregates in blood plasma of patients, (2) investigate whether patients with acute pulmonary embolism (PE) show blood plasma aggregates, and (3) test if treatment with an anticoagulant reduces blood plasma aggregates. We find that blood plasma aggregates are also present in patients with acute PE, that they reduce after treatment with enoxaparin, and that double centrifugation of plasma does not reduce aggregates. Finally, we raise important questions about the nature and pathophysiological role of aggregates.

## Methods
**Patients**. We collected plasma samples (pre-pandemic; 2018) from 35 patients with acute PE (aged $63.1 \pm 15.8$ years, 34% male) on admission to hospital and after 24 h of treatment with enoxaparin, a low-molecular weight heparin (LMWH) at a dose of 1.0 mg/kg s.c. every 12 h. Patients with known cancer, sepsis, autoimmune diseases, hypotension and pregnant women were excluded. All patients provided their written informed consent in accordance with the Declaration of Helsinki, and the Jagiellonian University Ethical Committee approved the study protocol (decision no. 122.6120.243.2015). Median fibrinogen concentration in this group was 3.4 (2.9–4.6) g/l.

**Blood sampling**. Venous blood was drawn without stasis onto 3.2% sodium citrate tubes (1 part of citrate to 9 parts of blood; SARSTEDT AG&Co., Nümbrecht, Germany), and blood samples were centrifuged at 2500×g for 10 min within 30 min to prepare platelet-poor plasma. In a follow-up study, plasma was taken from a patient with PE prior to treatment with LMWH, which was subsequently centrifuged twice at 2500 g for 10 min each time. All plasma samples were aliquoted, immediately frozen, stored at −80 °C, and shipped to the laboratory in Leeds on dry ice. Plasma samples were thawed in a 37 °C water bath for 10 min.

**Laser scanning confocal microscopy**. For laser scanning confocal microscopy, plasma was diluted 1/6 in tris-buffered saline, mixed with 50 μg/ml AlexaFluor488-labelled human fibrinogen, 0.1 U/ml human alpha thrombin (Merck, Millipore, Watford, UK) and 10 mM $CaCl_2$ (final concentrations), and loaded into the channel of an uncoated μ-slide (Ibidi GmbH, Gräfelfing, Germany). Plasma clots were allowed to form for 2 h in dark humidity chambers followed by imaging using a Zeiss LSM880 with 40× and 63× oil immersion objectives (Carl Zeiss, Welwyn Carden City, UK). To determine if the aggregates contained cell-membrane or other hydrophobic structures, DiOC6(3) (3,3′-Dihexyloxacarbocyanine Iodide, Thermo Fisher, 1.2 μg/ml) was added prior to the addition of thrombin. To determine if aggregates were present prior to clotting, the same protocol was followed without adding thrombin. We collected z-stacks (20 μm, 30 slices) and images were projected to a single plane followed by analysis using ImageJ (Wayne Rasband, NIH).

**Scanning electron microscopy**. For scanning electron microscopy, clots were formed by adding 1 volume activation mix (10 mM $CaCl_2$, 0.1 U/ml human alpha thrombin; final concentrations) to 10 volumes of plasma. Clots were allowed to form for 2 h in a humidity chamber at room temperature. After clotting, samples were rinsed, fixed using 2% glutaraldehyde, dehydrated, critical point dried, and sputter coated with iridium prior to imaging using a Hitachi SU8230 high-performance cold emission SEM (Chiyoda, Japan).

**Statistics and reproducibility**. We analysed the presence of aggregates in a total of 35 patients, both before and after treatment with LMWH. To determine the range of sizes of aggregates, we analysed a total of 49 aggregates from 9 plasma clots. Differences in categorical clinical parameters (e.g. gender, presence of other disease, drug treatments etc.) between patients with and without plasma aggregates were determined by Chi Square Testing. Differences in continuous clinical parameters (e.g. age, BMI, cell counts, lipids, protein concentrations etc.) between patients with and without plasma aggregates were determined by independent samples t-Tests. Throughout, $p$-values less than 0.05 were considered to indicate significance.

**Reporting summary**. Further information on research design is available in the Nature Portfolio Reporting Summary linked to this article.

## Results and discussion
Here we report the presence of aggregates in the blood plasma from patients with acute PE. We observed a striking presence of aggregates embedded within clots made from plasma of patients with acute PE. These aggregates were strongly positive for fluorescent fibrinogen, and some were positive for DiOC6(3) (Fig. 1). The surface area of the aggregates ranged from 61 to 3017 μm² ($n = 49$). Aggregates were also observed by scanning electron microscopy (Fig. 2). We, and others, normally observe a homogeneous fibrin fibre network in clots made from plasma of healthy individuals[5–7]. When fibrinogen is converted into fibrin by thrombin-mediated cleavage of the fibrinopeptides, fibrin forms a gel by polymerising into a random, three-dimensional network of branching fibrin fibres[8]. The presence of embedded aggregates within the clots from patients with PE suggested that the aggregates were already present in the plasma prior to clotting. We thus performed confocal microscopy on samples without thrombin addition and confirmed the presence of aggregates in the plasma (Fig. 1a–c). Aggregates also remained present in the plasma of a patient with acute PE after double centrifugation of

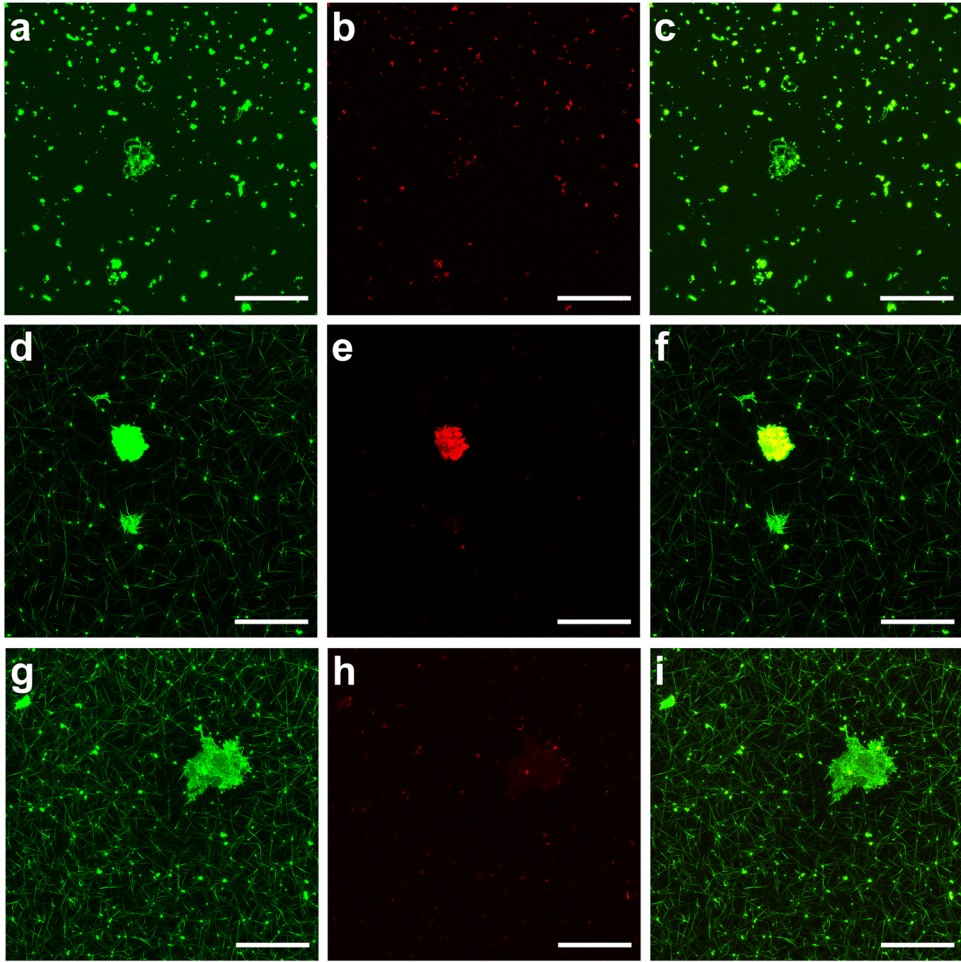

**Fig. 1 Laser scanning confocal microscopy images of aggregates in plasma and plasma clots from patients with PE.** Presence of aggregates in PE patient plasma prior to clot initiation with thrombin (**a–c**). Representative images of aggregates in PE patient plasma samples following the addition of thrombin showing platelet rich (**d–f**) and platelet poor (**g, h**) aggregates. Alexa Fluor 488 labelled fibrinogen is shown green (**a, d, g**) and DiOC6(3) staining is shown in red (**b, e, h**). An overlay of the two channels is also shown (**c, f, i**). Scale bars are 50 μm.

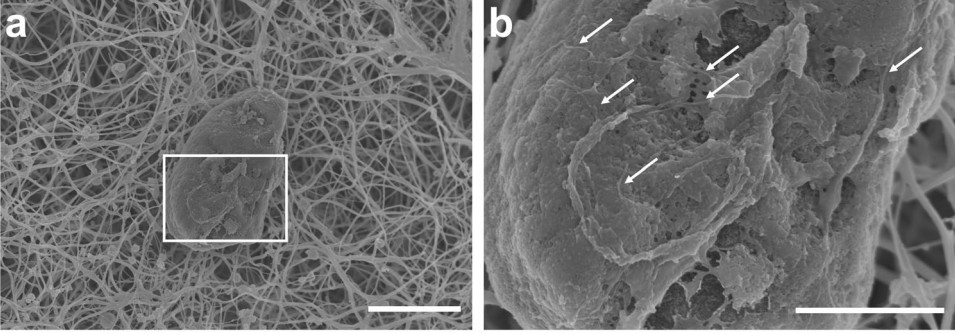

**Fig. 2 High-resolution scanning electron micrographs of an aggregate in a PE patient plasma clot.** The presence of aggregates was observed in clots using SEM (**a**). Small fibres can be seen within the aggregate (white arrows; **b**). Scale bars are 5 μm (**a**) and 2 μm (**b**) respectively.

the plasma, both before and after clotting with thrombin (Fig. 3). Aggregates were found in 8 (23%) out of 35 plasma clots from patients with acute PE. There were no differences in demographic and clinical variables, along with plasma fibrinogen, when the 8 patients were compared with the remainder. Strikingly, the presence of aggregates was substantially reduced in patients with acute PE receiving therapeutic-dose LMWH. After treatment, aggregates were found in only 1 (3%) out of 35 plasma clots. It thus appears that aggregates wain after acute PE, and/or

anticoagulation with LMWH reduced the presence of aggregates in plasma from patients with PE.

When we first observed aggregates in plasma from patients with acute PE, we were sceptical about our finding. We questioned whether aggregates were an artefact of blood sampling or plasma preparation procedures. However, our phlebotomists and laboratory staff preparing the samples are highly skilled, and aggregates remained present after double centrifugation of the plasma. Furthermore, we see aggregates in samples from patients

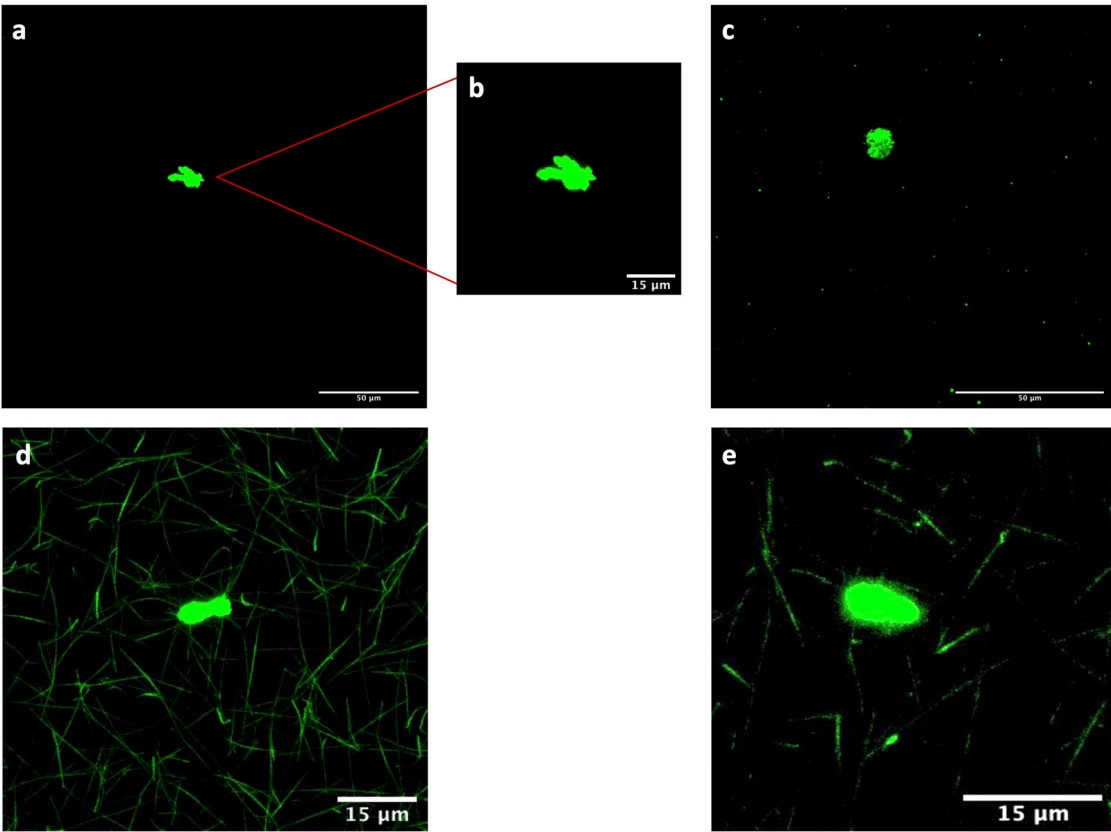

**Fig. 3 Aggregates in plasma and plasma clots from a PE patient after double centrifugation.** Blood samples were centrifuged twice in succession at 2500×$g$ for 10 min each time. Aggregates were present in the double-centrifuged plasma prior to clotting (**a**–**c**). Zoom-in on aggregate of around 15 µm in length (surface area ~175 µm$^2$) (**b**). Aggregates embedded in the fibrin network after clotting with thrombin and calcium (**d**, **e**). Alexa Fluor 488 labelled fibrinogen is shown green. Scale bars are 50 µm in **a** and **c** and 15 µm in **b**, **d**, and **e**.

with acute PE, but not in patients after LMWH treatment, thus suggesting anticoagulation reduces aggregate occurrence. An additional key argument in this regard is that we have not previously observed aggregates in plasma clots from healthy individuals[5–7]. Finally, these aggregates appear similar in structure and size as those recently reported in patients with diseases that associate with an increased risk of venous thromboembolism (VTE), i.e. type 2 diabetes and COVID-19[2–4], further making it less likely that this is an artefact.

Other key questions for future studies are elicited by the size of the aggregates. If the aggregates were present in the blood before the draw, why do they not precipitate into the pellet alongside the red blood cells (~50 µm$^2$ surface area) and platelets (~3–5 µm$^2$) during centrifugation for plasma preparation? Are they smaller to begin with, and then grow in the plasma or the clotting sample, acting as a nucleus for fibrin deposition? If we find small aggregates in the plasma of patients with acute PE, are there potentially larger aggregates in the whole blood of these patients prior to centrifugation? In addition, the size of the aggregates raises important questions about their potential pathophysiological role. If present in the blood, any aggregate >50 µm$^2$ may in principle block small arterioles in tissues like the lung for example. However, we have so far no evidence that the aggregates presented here and in other reports also occur in circulation in these patients and cause thrombosis themselves, or if they somehow represent a consequence and thus a sequela of clinical thrombosis.

Finally, important questions are raised about the mechanisms that underpin generation of aggregates in plasma samples from patients with VTE. As discussed above, a number of protein constituents have been found by mass-spectrometric analysis of aggregates in one study[4]. This raises important questions regarding the mechanism of aggregate generation, which may or may not be related to fibrin deposition. Indeed, the molecular and/or cellular mechanisms that underpin aggregate generation and their relevance will need to be fully elucidated to assess whether this is an epiphenomenon, or a pathophysiological mechanism related to thrombus formation or fibrinolysis, before we attempt any clinical management that could target this phenomenon.

In conclusion, we present evidence for the presence of aggregates in plasma samples of patients with acute PE, which were reduced after treatment with LMWH. The molecular and/or cellular mechanisms and possible role in the pathophysiology of VTE or other diseases deserve further study.

### Data availability
Authors can confirm that all relevant data are included in the article. For all confocal and electron micrographs, see Supplementary Data 1. For statistical analysis, see Supplementary Data 2 and 3.

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

## Acknowledgements

The Ariëns group is supported by grants from the British Heart Foundation (RG/18/11/34036) and the Wellcome Trust (204951/B/16/Z). F.L.M. is supported by the Wellcome Trust (215861/Z/19/Z).

## Author contributions

S.R.B. collected data and co-wrote the manuscript. G.H. contributed to primary data collection. M.Z. and A.U. provided patient samples and critically reviewed the manuscript. G.A. collected data on double centrifuged samples. F.L.M. set up new protocols for the study and critically reviewed the manuscript. B.J.H. contributed to study design and manuscript revision. R.A.S.A. conceived the study and wrote the manuscript.

## Competing interests

The authors declare no competing interests.
