## [Peer Review File · Communications Medicine]

Reviewers' comments:

Reviewer #1 (Remarks to the Author):

Baker et al describe an interesting finding that potentially microclots, aggregates of fibrin with platelets, may be present in the circulation of patients with pulmonary embolism that after treatment of these patients disappear. Although the evidence is circumstantial, this may suggest that these microclots are present in the circulation of the patients. This reviewer is not convinced though as there is no proof of circulating microclots (for instance by withdrawing blood in fixative and demonstrating microclots). This manuscript is however clearly hypothesis generating and warrants discussion on how microclots may appear in plasma. I think every clotting specialist has seen citrate plasma samples with fibrin clots that were unexpected from the anticoagulant present in the withdrawal tubes.

Specific comments:

1. Did the authors really withdraw blood in tubes containing 0.1% sodium citrate (what does 'venous blood was drawn onto 0.1% sodium citrate tubes' mean?)? This is clearly insufficient, and normal citrate tubes contain 3.2% (or 3.8% in some countries) sodium citrate giving a final concentration of 0.32%.
2. Blood was centrifuged once which leaves a considerable number of platelets in the platelet-poor plasma. Did the authors quantify the number of platelets in the platelet-poor plasma? Did the authors check for presence of microclots in double centrifuged plasma?
3. The microclots in Figure 1 contain fluorescent fibrin(ogen) in a system without thrombin addition. Since the labeled fibrinogen was added to the plasma of the patients, does this not suggest that incorporation occurred in vitro and may be seen as an ex vivo artefact?
4. If the theory of the authors is correct, the microclots may be derived from in vivo clots as 'normal' cleanup of the thrombus. It would therefore be indicative of lysis of existing clots. Treatment of patients with LMWH prevents growth of the thrombi, but is not thought to improve lysis. So, the reduction on LMWH is an indication that ex vivo formation of the clots is reduced because of an extra anticoagulant in the plasma. To this reviewer's opinion this does not exclude a laboratory artefact (line 92). Please discuss.
5. The figures could contain arrows to help the reader identify relevant visual aspects, for instance the small fibres in the aggregate of Figure 2B.

Joost C.M. Meijers

Reviewer #2 (Remarks to the Author):

This is a great paper! Congratulations for repeating results from other research groups.

This will have far-reaching consequences for coagulation studies and treatment regimes.

Reviewer #3 (Remarks to the Author):

The paper is clear and well written.

This is an interesting paper produced by a well-recognized group.

Although I'm not an expert in the laboratory techniques, it strikes me that extensive additional validation would be required to make a firm conclusion that these microaggregates are not in some way related to ex vivo preanalytic changes resulting from the withdrawal, handling, processing, preparation, and analysis of the samples. The process of anticoagulating samples with citrate or EDTA to avoid clotting prior to analysis is highly artificial and I suspect could contribute to these observations.

It is possible that these observations are real and that they might contribute to the biology however I think a lot of additional validation would be required before this would be widely accepted. For example, intravital microscopy of relevant animal experiments demonstrating circulation of such microaggregates would be helpful.

The fact that the aggregates are reduced in number after anticoagulation supports the hypothesis these are somehow artefactual because rather than relying on sodium citrate to produce anticoagulation ex vivo you are using a heparin to produce anticoagulation in vivo, reducing the likelihood of artefactual changes associated with the ex vivo handling of the sample. It would seem logical to me that drawing anticoagulated blood from a patient who has received enoxaparin into a citrated tube is much less likely to cause artefactual changes than when blood with normal coagulant potential is drawn into those same tubes.

Re: Baker S et al. Plasma from patients with pulmonary embolism show aggregates that reduce after anticoagulation. *Communications Medicine* submission.

We are grateful to the editor and the reviewers for their insightful comments and for allowing us to resubmit a revised manuscript. We have now revised the manuscript to address these comments, including the addition of new experimental data, and provide a point-by-point response to the reviewers' comments below. We added a new author (Fraser L Macrae) who performed experiments with different whole blood fixation methods, and another new author (Ghadir Alkarothi) who performed microscopy analysis of the new plasma samples (Figure 2). Changes to the manuscript are highlighted in yellow. We believe that these revisions have significantly improved the manuscript, and hope that it will be considered suitable for publication in *Communications Medicine*.

Reviewer #1 (Remarks to the Author):

Baker et al describe an interesting finding that potentially microclots, aggregates of fibrin with platelets, may be present in the circulation of patients with pulmonary embolism that after treatment of these patients disappear. Although the evidence is circumstantial, this may suggest that these microclots are present in the circulation of the patients. This reviewer is not convinced though as there is no proof of circulating microclots (for instance by withdrawing blood in fixative and demonstrating microclots). This manuscript is however clearly hypothesis generating and warrants discussion on how microclots may appear in plasma. I think every clotting specialist has seen citrate plasma samples with fibrin clots that were unexpected from the anticoagulant present in the withdrawal tubes.

Response: We completely concur with the reviewer that the presence of aggregates in the plasma from patients does not necessarily mean that these aggregates are also present in the circulation of the patients, prior to blood withdrawal. For this reason, we have chosen to present our findings alongside a clear discussion on mechanistic and pathophysiological aspects, highlighting a number of key scientific questions that remain to be answered with regards to the presence of aggregates in the plasma samples, before any treatment aimed at eliminating such aggregates is attempted in patients. Unfortunately, some private clinics offer plasma exchange or apheresis treatments without any evidence that these treatments actually work, with patients in desperation attempting these treatments at very high private cost. We therefore believe that our manuscript is very important as it raises essential points from a basic science point of view about the in-vitro aggregates that deserve further study before any conclusion about their causality or treatment is being considered.

The reviewer raises an interesting suggestion of taking blood on fixative. We have worked on this aspect for some time but ultimately unsuccessfully. We started with a detailed literature search for ex-vivo blood fixation and then tried a number of experimental in-vitro conditions on normal blood, to see which worked best. Gluteraldehyde fixation of whole blood did not work as it clotted the sample. We then tried a range of PFA concentrations, which indicated that 4% PFA for one hour would work, by not clotting the samples and allowing for fixed cells to be separated from fixed plasma. We tried this protocol with the blood from a patient with acute PE and no LMWH, which worked to the point of plasma separation and storage. The sample was sent on dry ice to Leeds. However, after arrival in Leeds unfortunately the sample clotted when it was thawed out. We again found aggregates by microscopy, but as the sample clotted upon thawing we do not think that these data are reliable and therefore did not include them in the revised manuscript.

Specific comments:

1. Did the authors really withdraw blood in tubes containing 0.1% sodium citrate (what does 'venous blood was drawn onto 0.1% sodium citrate tubes' mean)? This is clearly insufficient, and normal citrate tubes contain 3.2% (or 3.8% in some countries) sodium citrate giving a final concentration of 0.32%.

Response: We thank the reviewer for pointing this out and apologise for the mistake, which has now been corrected in the revised manuscript.

2. Blood was centrifuged once which leaves a considerable number of platelets in the platelet-poor plasma. Did the authors quantify the number of platelets in the platelet-poor plasma? Did the authors check for presence of microclots in double centrifuged plasma?

Response: This is a second interesting suggestion from the reviewer to test whether an additional centrifugation step reduces the presence of aggregates in the plasma samples. We have performed this experiment and found that aggregates are still present after double-centrifugation of the sample. We have added this data to the revised manuscript (new figure 2). However, while this shows that double centrifugation does not eliminate aggregates, the basic questions about causality and provenience of the aggregates remain, which is clearly pointed out in the manuscript. We did not quantify platelets in our samples, but did not see large platelet numbers on the micrographs.

3. The microclots in Figure 1 contain fluorescent fibrin(ogen) in a system without thrombin addition. Since the labelled fibrinogen was added to the plasma of the patients, does this not suggest that incorporation occurred in vitro and may be seen as an ex vivo artefact?

Response: From our experience and extensive experimental data, we know that the fluorescent fibrinogen added to the samples is completely inert. It behaves as native fibrinogen and does not start clotting or aggregation by itself. It will only bind to existing (or forming) clots, where it will interact with activated platelets, and is incorporated into the fibrin fibre network that is being formed. We do not see any fluorescence accumulation outside areas of clotting both in-vitro as well as in-vivo (via intravital microscopy) using fluorescently labelled fibrinogen as marker for thrombosis (e.g. Duval C et al. ATVB 2016; 36(2): 308-16).

4. If the theory of the authors is correct, the microclots may be derived from in vivo clots as 'normal' clean-up of the thrombus. It would therefore be indicative of lysis of existing clots. Treatment of patients with LMWH prevents growth of the thrombi, but is not thought to improve lysis. So, the reduction on LMWH is an indication that ex vivo formation of the clots is reduced because of an extra anticoagulant in the plasma. To this reviewer's opinion this does not exclude a laboratory artefact (line 92). Please discuss.

Response: We completely concur with the reviewer that the aggregates found in the plasma samples may occur as a consequence of thrombosis (or the normal cleanup as phrased by the reviewer). We cannot exclude this, nor can we evidence that the aggregates occur in the circulation. We also agree that the reduction in aggregates with LMWH does not exclude ex-vivo formation, which is why we use the word "suggesting" in this context. These points are reflected in the discussion of the revised manuscript. We have changed (we see aggregates in samples from patients with acute PE, but not in patients after LMWH treatment) "thus suggesting aggregates were no laboratory artefact" to "thus suggesting anticoagulation reduces aggregate occurrence"

5. The figures could contain arrows to help the reader identify relevant visual aspects, for instance the small fibres in the aggregate of Figure 2B.

Response: Arrows have been added to highlight the small fibres in figure 2B

Joost C.M. Meijers

Reviewer #2 (Remarks to the Author):

This is a great paper! Congratulations for repeating results from other research groups.

This will have far-reaching consequences for coagulation studies and treatment regimes.

Response: thank you for the positive comments.

Reviewer #3 (Remarks to the Author):

The paper is clear and well written.

This is an interesting paper produced by a well-recognized group.

Although I'm not an expert in the laboratory techniques, it strikes me that extensive additional validation would be required to make a firm conclusion that these microaggregates are not in some way related to ex vivo preanalytic changes resulting from the withdrawal, handling, processing, preparation, and analysis of the samples. The process of anticoagulating samples with citrate or EDTA to avoid clotting prior to analysis is highly artificial and I suspect could contribute to these observations.

Response: We completely concur with the reviewer and have for this reason constructed the manuscript in such a way that we describe our observations of aggregates in the plasma samples and plasma clots, but then add important discussion regarding the provenience of the aggregates, what causes their formation and whether or not they are implicated in the course of pulmonary embolism. We would like to point out that further studies are required as to the nature of the aggregates before interpretations are made regarding causality for disease and treatment of disease. We think it is unlikely that anticoagulation with citrate or EDTA contributes to the formation of aggregates, as we do not see such aggregates in plasma samples with those same anticoagulants from healthy individuals or patients with other disease.

It is possible that these observations are real and that they might contribute to the biology however I think a lot of additional validation would be required before this would be widely accepted. For example, intravital microscopy of relevant animal experiments demonstrating circulation of such microaggregates would be helpful.

Response: We concur with the reviewer that in-vivo studies of any aggregate formation using intravital microscopy in suitable animal models should be the subject for future studies. This is something we aim to pursue going forward, but takes too much time to develop and perform robustly for this manuscript revision.

As a note of interest, we recently developed a new murine model of pulmonary embolism (Duval C, PNAS 2021, doi: 10.1073/pnas.2103226118), in which emboli volumes as determined by light sheet microscopy ranged from 100 to 1600 μm^3 (see panel C in copy of published figure below). This volumetric range of murine micro-emboli translates to thrombi surface areas ranging from $(\sqrt[3]{100\mu\text{m}^3})^2 = 21.5 \mu\text{m}^2$ to $(\sqrt[3]{1600\mu\text{m}^3})^2 = 136.8 \mu\text{m}^2$, which surprisingly overlaps with the lower end of the range of surface areas of the plasma and plasma clot aggregates found in our current study (i.e. 61 to 3017 μm^2 ; n=49). New studies will be required to further investigate the in-vivo relevance of aggregates, their provenience, mechanisms of formation and their role in thromboembolism.

From Duval C, PNAS 2021:

Fig. 6. Light sheet microscopy of clot emboli in the lungs of FGG3X and WT mice. Following injection of 100 μg AlexaFluor647-fibrinogen and injury to the inferior vena cava with 5% FeCl₃ for 3 min, perfusion fixation of the mice with PFA after 57 min, and injection of FITC-albumin in the circulation prior collection and clearing of the lungs, light sheet fluorescence microscopy imaging of the lungs (A) showed that the total emboli count per mouse (B) was significantly increased in FGG3X mice. The distribution of all the emboli volumes (C) showed that FGG3X mice produced significantly more emboli, irrespective of their size, than WT mice (Scale bar, 1 mm). $n = 8$; [▲] males, [•] females. The data are presented as mean \pm SEM and analyzed by Mann–Whitney U test (B) and Kolmogorov–Smirnov test (C); * $P < 0.05$, *** $P < 0.001$.

The fact that the aggregates are reduced in number after anticoagulation supports the hypothesis these are somehow artefactual because rather than relying on sodium citrate to produce anticoagulation ex vivo you are using a heparin to produce anticoagulation in vivo, reducing the likelihood of artefactual changes associated with the ex vivo handling of the sample. It would seem logical to me that drawing anticoagulated blood from a patient who has received enoxaparin into a citrated tube is much less likely to cause artefactual changes than when blood with normal coagulant potential is drawn into those same tubes.

Response: We agree with the reviewer that the fact that anticoagulation with LMWH reduced aggregates does not prove whether the aggregates are produced in-vitro or occur in-vivo. We have changed the revised text to reflect this, see also response to reviewer 1, comment 4.

REVIEWERS' COMMENTS:

Reviewer #1 (Remarks to the Author):

The authors have adequately dealt with my concerns. I thank the authors for their elaborate rebuttal.

Reviewer #3 (Remarks to the Author):

The authors have addressed the issues identified on review.